# Exposure of Child Maltreatment Leads to a Risk of Mental Illness and Poor Prognosis in Taiwan: A Nationwide Cohort Study from 2000 to 2015

**DOI:** 10.3390/ijerph19084803

**Published:** 2022-04-15

**Authors:** Shi-Hao Huang, Iau-Jin Lin, Pi-Ching Yu, Bing-Long Wang, Chi-Hsiang Chung, Yao-Ching Huang, Wu-Chien Chien, Chien-An Sun, Gwo-Jang Wu

**Affiliations:** 1Department of Medical Research, Tri-Service General Hospital, Taipei 11490, Taiwan; hklu2361@gmail.com (S.-H.H.); iaujinlin@gmail.com (I.-J.L.); g694810042@gmail.com (C.-H.C.); 2Graduate Institute of Medicine, National Defense Medical Center, Taipei 11490, Taiwan; yupichin1003@gmail.com; 3Cardiovascular Intensive Care Unit, Department of Critical Care Medicine, Far-Eastern Memorial Hospital, New Taipei City 10602, Taiwan; 4School of Public Health, National Defense Medical Center, Taipei 11490, Taiwan; billwang1203@gmail.com; 5Taiwanese Injury Prevention and Safety Promotion Association (TIPSPA), Taipei 11490, Taiwan; 6Department of Chemical Engineering and Biotechnology, National Taipei University of Technology (Taipei Tech), Taipei 10608, Taiwan; 7Graduate Institute of Life Sciences, National Defense Medical Center, Taipei 11490, Taiwan; 8Department of Public Health, College of Medicine, Fu-Jen Catholic University, New Taipei City 24206, Taiwan; 040866@mail.fju.edu.tw; 9Big Data Center, College of Medicine, Fu-Jen Catholic University, New Taipei City 24206, Taiwan; 10Department of Obstetrics and Gynecology, Tri-Service General Hospital, Taipei 11490, Taiwan

**Keywords:** child maltreatment, homicide, psychiatric disorders, suicide

## Abstract

Objective: To investigate whether children with maltreatment exposure were associated with the risk of psychiatric disorders, suicide, and death. Methods: A retrospective cohort study was conducted, with 1592 child maltreatment cases and 6368 comparison cohort (1:4) matched for gender, age, and index year, from the Longitudinal Generation Tracking Database (LGTD2000) sampled from Taiwan National Health Insurance Research Database (NHIRD) in 2000, backtracking between 2000–2015 in Taiwan. The stratified Cox regression model was used to compare the risk of developing a mental illness and poor prognosis during the 15 years of follow-up. Results: There were 473 in the cohort with child maltreatment (675.10 cases per 100,000 person years) and 1289 in the comparison cohort (453.82 cases per 100,000 person years) that developed mental illness and poor prognosis. The stratified Cox regression model revealed that the adjusted hazard ratio (HR) was 1.91 to 11.76 (*p* < 0.05) after adjusting for monthly income level, occupation, and CCI after violence. Conclusion: Exposure to child maltreatment is associated with the risk of psychiatric disorders, but not suicide or death. This finding could be a reminder for clinicians about the mental health problems in patients with child maltreatment.

## 1. Introduction

World Health Organization (WHO) defined child maltreatment as the abuse and neglect that occurs to children under 18 years of age [1]. It includes all types of physical and/or emotional ill-treatment, sexual abuse, neglect, negligence, and commercial or other exploitation, which results in actual or potential harm to the child’s health, survival, development, or dignity in the context of a relationship of responsibility, trust, or power [1]. Any abusive, violent, coercive, forceful, or threatening act or word inflicted by one member of a family or household on another can constitute domestic violence. Family and Intimate Partner Violence (IPV) occurs largely between family members and intimate partners [2]. One in three women has been a victim of physical and/or sexual violence by an intimate partner at some point in her lifetime [3]. In England and Wales, child protection registrations increased by 182% (328.7 per 100,000 children) between 1988 and 2016 [4]. An average of nearly five children die every day from maltreatment or neglect in America [5]. The Ministry of Health and Welfare in Taiwan has reported that 4000 to 20,000 children were maltreated or neglected annually between 2003 and 2019 [6]. A history of maternal childhood physical maltreatment increased the risk of offspring physical maltreatment and neglect [7]. The negative role of child maltreatment was associated with various emotional problems and impaired psychosocial functioning [8]. In addition, Charles (2016) widely reviewed papers of this field and concluded that early life stress such as sexual, physical, emotional maltreatment, and emotional neglect, leads to a very significant increase in risk in adulthood of mood and anxiety disorders, substance and alcohol abuse, and certain other medical disorders [9].

Mental illness, also called mental health disorders, refers to a wide range of mental health conditions—disorders that affect your mood, thinking, and behavior. Examples of mental illness include depression, anxiety disorders, schizophrenia, eating disorders, and addictive behaviors [10]. Signs and symptoms of mental illness can vary, depending on the disorder, circumstances, and other factors, mental illness symptoms can affect emotions, thoughts, and behaviors [11]. Examples of signs and symptoms include: feeling sad or down, confused thinking or reduced ability to concentrate, excessive fears or worries, or extreme feelings of guilt, extreme mood changes of highs and lows, withdrawal from friends and activities, significant tiredness, low energy or problems sleeping, detachment from reality (delusions), paranoia or hallucinations, inability to cope with daily problems or stress, trouble understanding and relating to situations and to people, problems with alcohol or drug use, major changes in eating habits, sex drive changes, excessive anger, hostility or violence, suicidal thinking, and so on [10,11].

Although previous studies have pointed out that Adverse Childhood Experiences (ACE) will cause future psychiatric disorders [12], the association between child maltreatment and poor prognosis (including psychiatric disorders, suicide attempt, death) is yet to be investigated. Therefore, we hypothesized that child maltreatment victims would have a higher risk leading of mental illness and poor prognosis. We conducted this nationwide, population-based study to explore the risk of mental illness and poor prognosis in child maltreatment victims in Taiwan.

## 2. Method

### 2.1. Data Sources

This study used the inpatient and outpatient data of the Longitudinal Generation Tracking Database (LGTD2000) sampled of the Taiwan National Health Insurance (NHI) Research Database in 2000. All 2 million sampled people were tracked from 2000 to 2015 to explore the relationship between child maltreatment and mental illness and poor prognosis. The NHI program was established in 1995 and covers approximately 23 million residents in Taiwan. The population coverage of the NHI program was 99.0% and 99.5% in 2004 and 2010, respectively [13,14]. Before 2016, the International Classification of Diseases, 9th Revision, Clinical Modification (ICD-9-CM) was used for recording diagnosis in NHI Research Database [14]. The diagnosis of child maltreatment is recorded by the attending physician of the divisions of pediatrics, emergency, family medicine, and other related divisions based on clinical findings. Only psychiatrists made the psychiatric disorder diagnosis, judging by the Diagnostic and Statistical Manual of Mental Disorders, fourth edition (DSM-IV) [15], and the same edition of Text Revision (DSM-IV TR) [16]. Based on Article 50 of the Domestic Violence Prevention Act [17], Article 8 of the Sexual Assault Crime Prevention Act [18], and Articles 53 and 54 of the Protection of Children and Youth Welfare and Rights Act [19] in Taiwan, when physicians discovered that children or youth below the age of eighteen were being injured by domestic violence, the clinicians had to notify the competent authority under professional findings within 24 h. Thus, according to the high health insurance coverage and legally mandatory notification mechanism, the NHI Research Database is suitable for this study to investigate whether child maltreatment could be leading to mental illness and poor prognosis including psychiatric disorder, suicide attempt, or even death. The Ethical Review Board of the General Hospital of the National Defense Medical Center (C202105014) approved this study.

### 2.2. Study Design

We conducted a retrospective matched-cohort design. The ICD-9-CM code 995.50 was used to claim child maltreatment from 1 January 2000, to 31 December 2015. Victims under 18 years were enrolled as the child-maltreatment cohort (*n* = 1592). In addition, a 6368 matched comparison cohort with gender-, age-, CCI-, and index date-matched (1:4) were included without any child maltreatment experiences. The psychiatric disorders included anxiety, depression, manic disorder, bipolar, sleep disorders, post-traumatic stress disorder (PTSD), acute stress disorder (ASD), eating disorders, tobacco use disorder, alcoholism, alcohol abuse, drug dependence, drug abuse, schizophrenic disorders, and psychotic disorders. The victims with child maltreatment experiences or poor prognosis before the index date or year 2000 were excluded.

The covariates in this study contain gender, age, geographical area of residence (northern, central, southern, or eastern Taiwan), urbanization level (levels 1–4), level of the hospital (medical center, regional hospital, district hospital), and insurance premium category (in New Taiwan Dollars [NTD]; <18,000, 18,000–34,999, ≥35,000).

The levels of urbanization are categorized into seven stratifications according to population density (people/km^2^), the proportion of the population with a college or above education level (%), the proportion of people over the age of 65 (%), the proportion of the population classified as agricultural workers (%), and the number of physicians per 100,000 people in each city or county [20].

Comorbidities included Attention-Deficit Hyperactivity Disorder (ADHD), intellectual disability, autistic disorder/pervasive developmental disorder, conduct disorder/oppositional defiant disorder, other developmental disorders, childhood emotional disorder, Tourette syndrome/tics disorders, and enuresis/encopresis.

The Charlson comorbidity index (CCI) score with seventeen relevant comorbidities categories (based on the ICD-9-CM codes) was applied in this study [21]. The CCI score ranged from zero to thirty-seven, indicating no comorbidities through severe health issues.

The study tracking period is from 1 January 2000 to the onset of mental illness and poor prognosis, withdrawing from the NHI program or before the end of 2015. Mental illness and poor prognosis included suicide, death, and psychiatric disorders. In addition, affective psychosis (including anxiety, depression, manic disorder, bipolar), stress-related disorders (post-traumatic stress disorder/PTSD, acute stress disorder/ASD), sleep disorder (SD), eating disorders, substance-related disorders (including tobacco use disorder, alcoholism, alcohol abuse, drug dependence, and drug abuse), schizophrenic disorders, and psychotic disorders were classified as psychiatric disorders. Figure 1 shows the research-design flow chart of this study.

### 2.3. Statistical Analysis

These analyses were performed using SAS version 9.4 (SAS Institute, Cary, NC, USA). We use Generalized estimating equations (GEE) to evaluate the distribution of categorical and continuous variables. The Cox regression model was used to control the influence of covariates on the risk function of mental illness and poor prognosis, and the result is expressed as a Hazard Ratio (HR) of 95% CI. The Kaplan–Meier method was used to estimate the difference in risk leading to mental illness and poor prognosis between child maltreatment and matched cohorts. Two-tailed *p*-values less than 0.05 show a statistically significant difference.

## 3. Results

### 3.1. Characteristics of Study

The study included 1592 maltreated children and matched 6368 controls. The characteristics of the difference between the case and the control group at the beginning of the study are shown in Table 1. The average age of maltreated children was 13.9 ± 3.0 years, and the proportion of male children was 52.3%. The child maltreated cohort is different from the matched cohort in the following factors: insured premium; Charlson comorbidity index (CCI); location of the hospital; urbanization level; level of care. The mean following period was 8.05 (±6.89) and 8.14 (±6.92) years, respectively, in the child maltreated and matched cohort.

### 3.2. Risk of Mental Illness and Poor Prognosis According to Child Maltreatment Exposure

Abused children were linked to an increased risk of mental illness and poor prognosis (matched cohort as reference: adjusted HR [aHR] = 1.91 to 11.76; *p* < 0.05) (Table 2).

In the current study, compared with children who have not been violently maltreated, there is no increase in the risk of suicide or death (broadly) for children who have been violently maltreated (regardless of gender) but is only related to the increased risk of 13 mental illnesses. Including anxiety, depression, bipolar disorder, SDs, PTSD, ASD, tobacco use disorder, alcoholism, alcohol abuse, drug dependence, drug abuse, schizophrenic disorders, and psychotic disorders, except for mania and eating disorders. Among them, the three most risky mental diseases are drug abuse, drug dependence, and PTSD, and their hazard ratios are 11.76, 11.24, and 9.01, respectively (Table 2).

Among boys affected by violence, compared with boys who have not suffered violent maltreatment, the risk of developing eight mental illnesses is significantly increased, including anxiety, depression, bipolar disorder, SDs, ASD, and tobacco use disorder, alcoholism, drug dependence, except mania, PTSD, eating disorders, alcohol abuse, drug abuse, schizophrenic disorders, psychotic disorders. Among them, the three most risky mental diseases are drug dependence, ASD, and bipolar disorder, and their hazard ratios are 9.19, 4.95, and 3.92, respectively (Table 3).

Among girls affected by violence, compared with girls who have not suffered violence, the risk of developing 12 kinds of mental illnesses is significantly increased, including anxiety, depression, bipolar disorder, SDs, PTSD, ASD, alcoholism, and alcohol abuse, drug dependence, drug abuse, schizophrenic disorders, and psychotic disorders, except for mania, eating disorders, and tobacco use disorder. Among them, the three most risky mental diseases are drug abuse, PTSD, and drug dependence, and their hazard ratios are 26.2, 15.37, and 14.35, respectively (Table 4). The range of comorbidities occurrence in both groups see Appendix A for details.

## 4. Discussion

The result of the study revealed that child maltreatment was associated with more than a 1.803-fold increased risk of mental illness. The incidences of anxiety, depression, bipolar, SDs, PTSD, ASD, tobacco use disorder, alcoholism, alcohol abuse, drug dependence, drug abuse schizophrenic disorders, psychotic disorders were higher in children with abuse expose than persons with no maltreatment exposure. Among them, the three most dangerous mental disorders were drug abuse, drug dependence, and PTSD, with hazard ratios of 11.76, 11.24, and 9.01, respectively. Among boys and girls affected by violence, the three most risky mental diseases were drug dependence, ASD, and bipolar disorder, and their hazard ratios are 9.19, 4.95, and 3.92, respectively (Male); drug abuse, PTSD, and drug dependence, and their hazard ratios are 26.2, 15.37, and 14.35, respectively (Female). Therefore, child maltreated victims would have a higher risk leading to a mental illness and poor prognosis in Taiwan.

Suicide and mortality are the most harmful situation of psychiatric disorders. Patients with psychiatric disorders face a higher risk of suicide [22]. Previous research pointed out, persons exposed to child maltreatment have a considerable excess risk of death during late adolescence and young adulthood [23]. Our research results do not support that child suffering from violence leads to a higher risk of suicide and death.

Poor prognosis is a relatively broad term. As far as we know, this is the first attempt to integrate psychiatric disorders with suicide and death as a unit to examine the effects on children of maltreatment.

### 4.1. Comparison with Previous Observations

Victims of child maltreatment are at elevated risk for suicide in adolescence and young adults [23,24]. In our study, children under the age of 18 tend not to be given a diagnosis of suicide. Therefore, it may underestimate the suicide situation in children. However, based on the fact that the child who has been violently maltreated and the control group have been accurately matched by gender, age, and CCI, With the same observation conditions, despite this, no suicide cases occurred in the control group.

Kisely et al. (2020) indicated that the lifetime incidence rates of anxiety disorder, depressive disorder, and PTSD were 43%, 22%, and 9%, respectively, over the 30 years follow-up. In addition, the adjusted ORs of anxiety disorder, depressive disorder, and PTSD were 1.41, 1.71, and 2.32, respectively, due to child maltreatment [25]. Gardner’s (2019) systematic reviews also pointed out the relations between child maltreatment and anxiety (OR 1.68, 95% CI 1.33–2.14), depression (OR 2.48, 95% CI 2.14–2.87), and PTSD (OR 3.35, 95% CI 1.55–7.22) [26]. Previous studies also support that bipolar [27], SDs [28], eating disorders [29], tobacco use disorders [30], alcoholism [31], and drug abuse [32] were associated with child maltreatment. The same results also could be observed in our study, except for eating disorders.

In our study, the relations between manic disorders, eating disorders with child maltreatment were not statistically significant. However, a previous study pointed out that there is a growing consensus in the field that experiences of child maltreatment contribute to the onset of psychotic symptoms and psychotic disorders [33]. The mechanisms by which experiences of child maltreatment confers risk for psychotic disorders remain unknown. Few studies have explored the difference in the risk of mental illness between boys and girls after being injured by violence. However, this study found that girls have a higher risk of mental illness after violence (12 types) than boys (8 types).

As Jaffee’s (2017) annual review mentioned, the master challenge is establishing a causal role for maltreatment and psychiatric disorders [34]. In this study, a retrospective cohort study was used to track the medical records of 99.7% of Taiwan’s population for 15 years, avoiding the common bias in cross-sectional studies. Compulsory notification regulations and the popularity of medical treatment in the entire population have increased the accuracy of the results of this study.

Previous studies provide evidence of the mechanism between child maltreatment and mental illness and poor prognosis. Aas’s (2019) study supports childhood trauma as an independent risk factor for bipolar [35]. A child’s early abuse negatively impacts the developmental trajectory of the right brain, dominant for attachment, affect regulation, and stress modulation, thereby setting a template for the coping deficits of both mind and body, therefore resulting in PTSD [36].

An early study explained why the maltreatment of children causes poor prognosis: (i) The experience of physical maltreatment in early childhood is a risk marker for the development of chronic aggressive behavior patterns. (ii) Harmed children are likely to develop biased and deficient patterns of processing social information [37].

The current prevalent genetic research attempts to explain the relationship between the maltreatment of children and mental illness. However, our study supports that maltreatment of children could lead to a mental illness and poor prognosis. Avoiding child maltreatment can prevent a mental illness and poor prognosis. More specific studies such as subgroup analysis and the influence of different age groups need to be conducted.

### 4.2. Limitations

The present study has several limitations that warrant consideration. First, similar to the previous study using the NHI research database on psychiatric disorders [12], we were unable to evaluate the genetic factor, psychosocial factor, environmental factor, severity, or psychological assessments in the patients with psychiatric disorders, since the data were not recorded in the NHI research database. Second, the subjects of this study are children who have been exposed to violence and seek medical care. Therefore, only those who have experienced more severe violence will be observed by the investigator, and the incidence may be underestimated.

## 5. Conclusions

The results showed that child maltreatment victims would have a higher risk leading to mental illness and poor prognosis in Taiwan. The three most dangerous mental disorders were drug abuse, drug dependence, and PTSD (Total); drug dependence, ASD, and bipolar disorder (Male); drug abuse, PTSD, and drug dependence (Female). Our findings support the strong association between maltreated children and the increased risk of psychiatric disorders. All people involved in caring for maltreated children, including family members, social workers, medical personnel, and legislators, should be aware that maltreated children are at higher risk of psychiatric disorders. For children who have suffered maltreatment, we should provide an early diagnosis of mental illness (especially in drug abuse, drug dependence, ASD, bipolar disorder, and PTSD) and psychological counseling assistance.

## Figures and Tables

**Figure 1 ijerph-19-04803-f001:**
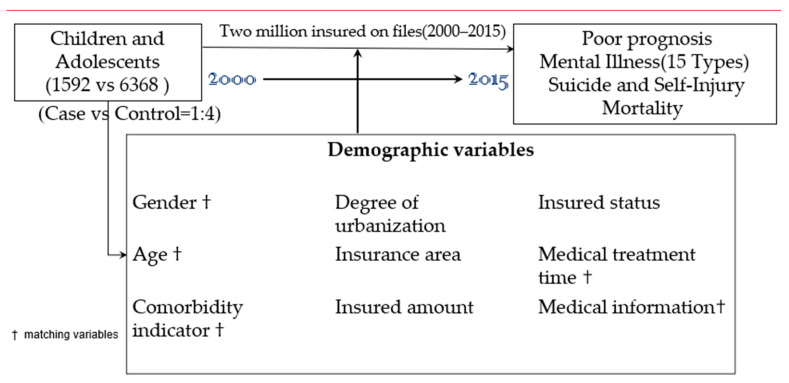
The flowchart of the study sample selection.

**Table 1 ijerph-19-04803-t001:** Basic demographic data on medical treatment of children and adolescent victims and the control group.

Demographic Variables		Child and Adolescent Victims (*n* = 1592)		Control Group (*n* = 6368)		*p*
		*n*	*%*	*n*	*%*	
Gender	Female	759	47.7	3036	47.7	>0.999
	Male	833	52.3	3332	52.3	
Age	Mean (SD)	13.9	(3.0)	13.9	(3.0)	0.4749
	Median (Min-Max)	15	(0–17)	15	(0–17)	
Urbanization	Highly urbanized town	265	16.6	1670	26.2	0.8145
	Moderately urbanized town	473	29.7	2035	32.0	
	New town	387	24.3	1194	18.8	
	General townships	247	15.5	844	13.3	
	Aging towns + Remote towns + Outlying islands	139	8.7	347	5.4	
	Agricultural township	74	4.6	231	3.6	
	Missing value	7	0.4	47	0.7	
Insurance Area	Taipei	394	24.7	2249	35.3	0.8451
	North District	223	14.0	1102	17.3	
	Central District	483	30.3	1054	16.6	
	South area	181	11.4	835	13.1	
	Kaohsiung and Pingtung	264	16.6	899	14.1	
	Eastern District	41	2.6	185	2.9	
	Missing value	6	0.4	44	0.7	
Insured Amount	0–16,499 (TWD)	511	32.1	1424	22.4	<0.001 **
	16,500–20,999 (TWD)	475	29.8	1893	29.7	
	21,000–30,299 (TWD)	361	22.7	1507	23.7	
	≥30,300 (TWD)	245	15.4	1544	24.2	
Personal or Health insurance	Public insurance	36	2.3	361	5.7	<0.001 **
dependent occupation	labor protection	822	51.6	3869	60.8	
	Farmer	187	11.7	612	9.6	
	Member of Water Conservancy and Fisheries Association	52	3.3	185	2.9	
	low-income households	133	8.4	168	2.6	
	Community insured population	346	21.7	1119	17.6	
	Other + missing values	16	1.0	54	0.8	

Tested by GEE, correlation matrix: unstructured, **: *p* < 0.01; other occupations for the person or the health care dependent include religious persons, placement in other social welfare institutions, veterans, others, and missing values.

**Table 2 ijerph-19-04803-t002:** Comparison of the risk of mental illness and poor prognosis between victims of violence and those without violence (children and adolescent).

Poor Prognosis	Violent Injurer (*n* = 1592)	Not Violent Injurer (*n* = 6368)	*HR (95% CI)*	*p*
	Incidence Rate (1/10^4^)	Incidence Rate (1/10^4^)		
Psychiatric comorbidity				
Anxiety	164.7	61.9	2.41 (1.97–2.94)	<0.0001 **
Depression	114.7	32.1	3.23 (2.51–4.17)	<0.0001 **
Manic disorder	2.3	0.4	5.36 (0.89–32.21)	0.0666
Bipolar disorder	29.0	4.1	6.54 (3.56–12.04)	<0.0001 **
Sleep disorder	264.1	118.6	1.91 (1.64–2.23)	<0.0001 **
Posttraumatic stress disorder	16.5	1.8	9.01 (3.94–20.62)	<0.0001 **
Acute stress disorder	25.9	4.7	5.42 (2.98–9.84)	<0.0001 **
Eating disorders	5.5	3.7	1.68 (0.66–4.25)	0.2727
Tobacco use disorder	40.8	13.1	2.58 (1.72–3.85)	<0.0001 **
Alcoholism	15.6	4.5	2.94 (1.53–5.67)	0.0012 **
Alcohol abuse	11.7	2.5	4.36 (1.89–10.05)	0.0006 **
Drug dependence	14.1	1.0	11.24 (4.11–30.69)	<0.0001 **
Drug abuse	9.4	1.0	11.76 (3.24–42.74)	0.0002 **
Schizophrenic disorders	8.6	4.1	2.18 (1.00–4.73)	0.0495 *
Psychotic disorders	16.4	3.3	4.47 (2.26–8.84)	<0.0001 **
Suicide and self-inflicted injury	4.7	0	-	-
Mortality	16.4	13.5	1.0 (0.58–1.72)	0.9963

Using stratified Cox regression analysis, the CCI, insured amount, insured identity, gender, age, and pre-matching CCI have been corrected at the time of pairing. *: *p* < 0.05, **: *p* < 0.01.

**Table 3 ijerph-19-04803-t003:** Comparison of the risk of mental illness and poor prognosis between victims of violence and those without violence (children and adolescent, male).

Poor Prognosis	Violent Injurer (*n* = 833)				Not violent Injurer (*n* = 3332)				*HR (95% CI)*	*p*
	*n*	%	Total Person Years	Incidence Rate (1/10^4^)	*n*	%	Total Person Years	Incidence Rate (1/10^4^)		
Psychiatric comorbidity										
Anxiety	94	11.3	7240	129.8	142	4.3	26,522	53.5	2.14 (1.59–2.88)	<0.0001 **
Depression	53	6.4	7296	72.6	67	2.0	26,617	25.2	2.27 (1.51–3.43)	<0.0001 **
Manic disorder			7434				26,714		(-)	
Bipolar disorder	11	1.3	7418	14.8	8	0.2	26,714	3.0	3.92 (1.42–10.85)	0.0084 **
Sleep disorder	151	18.1	7239	208.6	253	7.6	26,433	95.7	1.86 (1.49–2.34)	<0.0001 **
Posttraumatic stress disorder	≤5			4.1	≤5			1.5	2.71 (0.60–12.25)	0.1947
Acute stress disorder	12	1.4	7389	16.2	11	0.3	26,686	4.1	4.95 (1.98–12.41)	0.0006 **
Eating disorders	≤5			1.3	≤5			1.5	1.00 (0.11–8.95)	0.9999
Tobacco use disorder	43	5.2	7381	58.3	55	1.7	26,702	20.6	2.62 (1.70–4.03)	<0.0001 **
Alcoholism	11	1.3	7413	14.8	13	0.4	26,691	4.9	2.65 (1.08–6.51)	0.0341 *
Alcohol abuse	7	0.8	7421	9.4	8	0.2	26,714	3.0	3.16 (0.96–10.38)	0.0576
Drug dependence	10	1.2	7410	13.5	≤5			1.1	9.19 (2.44–34.59)	0.0010 **
Drug abuse	≤5			5.4	≤5			1.1	4.89 (0.85–28.26)	0.0762
Schizophrenic disorders	≤5			5.4	15	0.5	26,700	5.6	0.98 (0.31–3.11)	0.9768
Psychotic disorders	7	0.8	7431	9.4	10	0.3	26,705	3.7	1.87 (0.66–5.30)	0.2370
Suicide and self-inflicted injury			7434				26,714		(-)	
Mortality	18	2.2	7434	24.2	47	1.4	26,714	17.6	1.10 (0.61–1.97)	0.7566

Using stratified Cox regression analysis, the CCI, insured amount, insured identity, gender, age, and pre-matching CCI have been corrected at the time of pairing. -: The model does not converge, and the reference group is the unexposed group. *: *p* < 0.05, **: *p* < 0.01.

**Table 4 ijerph-19-04803-t004:** Comparison of the risk of mental illness and poor prognosis between victims of violence and those without violence (children and adolescent, female).

Poor Prognosis	Violent Injurer (*n* = 759)				Not Violent Injurer (*n* = 3036)				*HR (95% CI)*	*p*
	*n*	%	Total Person Years	Incidence Rate (1/10^4^)	*n*	%	Total Person Years	Incidence Rate (1/10^4^)		
Psychiatric comorbidity										
Anxiety	109	14.4	5088	214.2	157	5.2	21,762	72.1	2.67 (2.03–3.50)	<0.0001 **
Depression	89	11.7	5089	174.9	89	2.9	21,924	40.6	4.05 (2.93–5.61)	<0.0001 **
Manic disorder	≤5			5.6	≤5			0.9	5.36 (0.89–32.21)	0.0666
Bipolar disorder	26	3.4	5328	48.8	12	0.4	22,072	5.4	8.62 (3.95–18.8)	<0.0001 **
Sleep disorder	173	22.8	5030	343.9	318	10.5	21,716	146.4	1.95 (1.59–2.41)	<0.0001 **
Posttraumatic stress disorder	18	2.4	5324	33.8	≤5			2.3	15.37 (5.14–46.02)	<0.0001 **
Acute stress disorder	21	2.8	5340	39.3	12	0.4	22,051	5.4	5.78 (2.63–12.7)	<0.0001 **
Eating disorders	6	0.8	5384	11.1	14	0.5	22,038	6.4	1.93 (0.68–5.45)	0.2160
Tobacco use disorder	9	1.2	5367	16.8	9	0.3	22,072	4.1	2.34 (0.76–7.15)	0.1374
Alcoholism	9	1.2	5371	16.8	9	0.3	22,059	4.1	3.32 (1.28–8.63)	0.0137 *
Alcohol abuse	8	1.1	5384	14.9	≤5			1.8	6.04 (1.78–20.53)	0.0039 **
Drug dependence	8	1.1	5380	14.9	≤5			0.9	14.35 (3.04–67.71)	0.0008 **
Drug abuse	8	1.1	5362	14.9	≤5			0.9	26.2 (3.22–213.27)	0.0023 **
Schizophrenic disorders	7	0.9	5361	13.1	≤5			2.3	6.19 (1.78–21.44)	0.0041 **
Psychotic disorders	14	1.8	5336	26.2	6	0.2	22,081	2.7	9.80 (3.50–27.49)	<0.0001 **
Suicide and self-inflicted injury	6	0.8	5368	11.2	0	0.0	22,085	0.0	-	-
Mortality	≤5			5.6	19	0.6	22,085	8.6	0.61 (0.14–2.73)	0.5184

Using stratified Cox regression analysis, the CCI, insured amount, insured identity, gender, age, and pre-matching CCI have been corrected at the time of pairing. -: The model does not converge, and the reference group is the unexposed group. *: *p* < 0.05, **: *p* < 0.01.

## Data Availability

Data are available from the National Health Insurance Research Database (NHIRD) published by the Taiwan National Health Insurance (NHI) Administration. Due to legal restrictions imposed by the government of Taiwan concerning the “Personal Information Protection Act”, data cannot be made publicly available. Requests for data can be sent as a formal proposal to the NHIRD (https://dep.mohw.gov.tw/DOS/lp-2506-113.html accessed on 13 March 2022).

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
