# Peer review of "Exposure of Child Maltreatment Leads to a Risk of Mental Illness and Poor Prognosis in Taiwan: A Nationwide Cohort Study from 2000 to 2015"

_ijerph, 2022, doi:10.3390/ijerph19084803_

Round 1

Reviewer 1 Report

The topic is extremely relevant, and is especially important in the work of specialist psychiatrists in intervening with families, enabling the distinction of pathological from clinically relevant.

the introduction is objective

Materials and methods:The sample is large enough, statistical analysis is adequate

Critically, it’s important to establish what is novel about this study, such as the Taiwan population being studied (and why) or to test some specific relationship that might explain why it was important to take both state and trait measures of anxiety

Results: they are presented correctly and with relevant indicators

Key words: MeSH indexed key words should be used. Order them alphabetically.

please pay attention to English grammar

Reviewer 2 Report

The researchers take up an important social problem: assessing the relationship between specific disorders and health problems in children and adolescents experiencing maltreatment. This problem is verified in the socio-cultural realities of Taiwan, which should be considered when generalizing the obtained trends. The issues discussed in the article align with the thematic profile of the journal. Appreciating the importance of the issue from the cognitive and practical perspective, I want to point out some shortcomings that, in my opinion, should be eliminated in order to improve the manuscript and ensure its adequate scientific value:

  1. The Authors should ensure that terminology is standardized. The article includes a number of terms whose meaning can hardly be considered the same (maltreatment, abuse, violence, negligence). Explanations and precise indicators are missing, especially in the context of the Authors’ own research.
  2. Please specify whether the analyzed cases were confirmed or only “suspected” cases (p. 2, 79-81).
  3. I think it is worth clarifying, as long as these were confirmed cases, what kind of domestic violence the children experienced. This may potentially differentiate between short-term and long-term health consequences.
  4. The Authors analyzed comorbidities in both groups. Please provide data showing the range of their occurrence. This is a very important variable that can differentiate both the extent of domestic violence, its form, and its consequences.
  5. Research into this issue is important from a clinical, pedagogical, and psychological perspective. Please, provide more detailed implications, since now they are only very vaguely formulated in the conclusions.
